# Targeted Resequencing of Otosclerosis Patients from Different Populations Replicates Results from a Previous Genome-Wide Association Study

**DOI:** 10.3390/jcm11236978

**Published:** 2022-11-26

**Authors:** Lisse J. M. Tavernier, Thomas Vanpoucke, Isabelle Schrauwen, Guy Van Camp, Erik Fransen

**Affiliations:** 1Center of Medical Genetics, University of Antwerp & Antwerp University Hospital, 2650 Antwerp, Belgium; 2Center for Statistical Genetics, Department of Neurology, Gertrude H. Sergievsky Center, Columbia University Medical Center, New York, NY 10032, USA; 3StatUA Center for Statistics, University of Antwerp, 2610 Antwerp, Belgium

**Keywords:** otosclerosis, hearing loss, targeted resequencing, replication, gene analysis

## Abstract

Otosclerosis is one of the most common causes of hearing loss in young adults. It has a prevalence of 0.3–0.4% in the European population. Clinical symptoms usually occur between the second and fifth decade of life. Different studies have been performed to unravel the genetic architecture of the disease. Recently, a genome-wide association study (GWAS) identified 15 novel risk loci and replicated the regions of three previously reported candidate genes. In this study, seven candidate genes from the GWAS were resequenced using single molecule molecular inversion probes (smMIPs). smMIPs were used to capture the exonic regions and the 3′ and 5′ untranslated regions (UTR). Discovered variants were tested for association with the disease using single variant and gene-based association analysis. The single variant results showed that 13 significant variants were associated with otosclerosis. Associated variants were found in five of the seven genes studied here, including *AHSG*, *LINC01482*, *MARK3*, *SUPT3H* and *RELN*. Conversely, burden testing did not show a major role of rare variants in the disease. In conclusion, this study was able to replicate five out of seven candidate genes reported in the previous GWAS. This association is likely mainly driven by common variants.

## 1. Introduction

Otosclerosis is one of the most common causes of hearing loss in young adults. The disease is caused by an abnormal bone remodeling in the middle and inner ear, where normal bone is replaced by otosclerotic bone [1]. The bone formation can lead to fixation of the stapes, the other ossicles or the round window membrane [2]. Clinical signs of otosclerosis usually start between the second and fifth decade of life and present mostly as a conductive hearing loss, although sensorineural or mixed hearing loss have also been reported [3,4]. Clinical otosclerosis has a prevalence of 0.3–0.4% in the European population [5].

Otosclerosis occurs in individuals with a large family history as well as in sporadic cases, where no family history is known. In large families, the disease is inherited as an autosomal dominant trait with reduced penetrance [6,7]. Linkage analysis in these families has led to the identification of 8 different loci [8,9,10,11,12,13,14,15]. However, resequencing of candidate genes has never led to the identification of a monogenic disease-causing variant or gene. Sporadic cases are probably attributable to a combination of genetic and environmental factors. Several association studies have been performed, where analysis of common variants in functional candidate genes identified associated variants in different genes such as *TGFβ1*, *RELN*, *COL1A1*, *BMP2* and *BMP4*. Targeted resequencing of candidate genes has identified new pathogenic variants in genes as *MEPE* and *ACAN* [16,17].

The disadvantage of the previously mentioned association studies is that association of variants with otosclerosis is studied in predetermined genes. To avoid this, genome-wide association studies (GWAS) can be performed to study thousands of common variants throughout the genome. This allows us to test all genes across the genome in a hypothesis-free way. To date, only two GWA studies have been performed in otosclerosis. The most recent GWAS [18] was performed as a meta-analysis from three different biobanks: FinnGen, EstBB and UKBB, resulting in the identification of 18 loci associated with otosclerosis, including genes that were previously associated with otosclerosis, e.g., *RELN*,* TGFβ1* and *MEPE*. In addition, 15 novel loci were identified, harboring several genes with an important role in regulation of the osteoblast and osteoclast, in bone mineralization or in various skeletal disorders. This GWAS was an important breakthrough in otosclerosis research.

One of the keys to the elucidation of a complex disease is the replication of association results in multiple independent populations. The goal of the present study is to replicate the results from the GWAS, focusing on seven of the 18 identified genes, and to identify additional variants within each gene associated with otosclerosis.

## 2. Materials and Methods

### 2.1. Selection of Cases and Controls

Patients and controls, collected for the purpose of genetic research in otosclerosis and used in previous studies, were used and no additional samples were collected for this current study. The cohort was used in the past to study associations of different genes with otosclerosis (e.g., *TGFβ1*, *BMP2*, *BMP4*, *RELN* and *COL1A1*) [19,20,21,22] and for targeted resequencing of *MEPE*, *SERPINF1* and *ACAN* [16,17,23].

Patients were diagnosed with otosclerosis based either on surgical findings during stapes microsurgery or on a combination of audiological and clinical data. These data consist of medical history, otoscopy, tympanometry, acoustic reflex testing and audiometry. Measurements of pure tone audiometry (at 125, 250, 500, 1000, 2000, 4000 and 8000 Hz) and bone conduction (at 250, 500, 1000, 2000 and 4000 Hz) were performed. Fixation and mobility of the stapes was determined by tympanometry and stapedial reflexes.

Collection of patients and controls was done in eight centers, namely at Center of Medical Genetics, University Hospital of Antwerp (Edegem, Belgium), Department of Clinical Sciences and Community Health, University of Milan (Milan, Italy), Jean Causse Ear Clinic (Colombiers, France), ENT Department, Iuliu Hatieganu University of Medicine and Pharmacy (Cluj-Napoca, Romania), Department of Otolaryngology, University Hospital of Antwerp (Antwerp, Belgium), GZA Hospital campus Sint-August (Antwerp, Belgium), University Hospital of Ghent (Ghent, Belgium), Radboud University Medical Center (Nijmegen, the Netherlands) and University Hospital Zurich (Zurich, Switzerland). Patients include both sporadic and familial cases, whereby only one member of each family was selected for inclusion to ensure that all selected cases were unrelated to each other. Controls were matched to cases based on age, gender and ethnicity.

Informed consent was given by all participants prior to the study. This study was approved by the Ethical Committee of the University of Antwerp (UA A10-07) and all procedures were approved by local ethics committees. Approval was given in accordance with the World Medicals Association’s Declaration of Helsinki.

### 2.2. Gene Selection and Targeted Enrichment

Seven genes were selected for resequencing based on the results of the GWAS by Rämö et al. [18]. The replication effort prioritized (i) variants showing the strongest association with otosclerosis in the GWAS, (ii) variants in exonic regions, and (iii) variants in small genes, increasing the number of distinct genes that could be included in the study. For all genes single molecule molecular inversion probes (smMIPs) were designed to cover the coding region and 3′ and 5′ untranslated regions (UTR) with an overhang of the exon-intron boundaries of at least 10 bases. 224 smMIPs were designed using MIPgen [24] and ordered from IDT (Integrated DNA Technologies) (Coralville, IA, USA). smMIP enrichment was performed similarly to previous described protocols [16,17,25].

A pool of all 224 smMIPs was phosphorylated for 45 min at 37 °C with T4 Polynucleotide Kinase (New England Biolabs, Ipswich, MA, USA), H_2_O and 10x T4 DNA ligase buffer with 10 mM ATP (New England Biolabs, Ipswich, MA, USA). The phosphorylated pool was further diluted in EB buffer (Qiagen, Hilden, Germany) to reach a genomic DNA to smMIP ratio of 1 to 800. smMIP capture was performed in 5 µL (20 ng/µL) genomic DNA. Afterwards, remaining linear DNA was digested using 20,000 U/mL Exonuclease I (*E. coli*) (New England Biolabs, Ipswich, MA, USA) and 100,000 U/mL Exonuclease III (*E. coli*) (New England Biolabs, Ipswich, MA, USA). Captured DNA was amplified with PCR, using standard techniques. In total ten different barcoded forward primers and 384 different barcoded reverse primers were combined to ensure a unique barcode combination for each sample. All samples were pooled and purified using Agencourt AMPureXP magnetic beads (Beckman Coulter Life Sciences, Indianapolis, IN, USA). Purified pools were diluted to 2 nM and sequenced in three runs on the Nextseq 500 using the High Output Kit v2.5 (300 Cycles) (Illumina, San Diego, CA, USA).

### 2.3. Variant Calling and Statistical Testing

Using an in-house bioinformatics pipeline, VCF files were generated from the obtained sequencing data. FASTQ files of all runs were merged and samples were demultiplexed utilizing the unique combination of indexes. Illumina adaptors and low quality bases were trimmed prior to alignment to the hg19 reference genome using the BWA-MEM alignment tool version 0.7.17 (Li H., https://bio-bwa.sourceforge.net/index.shtml, accessed on 23 November 2022) [26]. PCR duplicates were removed based on the random eight base pair nucleotide tag of the smMIPs. Variant calling was performed for each sample separately using GATK haplotypecaller v.4.0.3.0 (Broad Institute, https://gatk.broadinstitute.org/hc/en-us, accessed on 23 November 2022) [27] with a minimum base quality of 35, a quality by depth of 3.89 and an allele depth cutoff of 5. In addition, genotype calling was adjusted based on the fraction of alternative allele in the reads. Variants were called homozygous reference when the alternative allele was present in less than 20% of the reads, heterozygous when the alternative allele was present in 20% to 80% of the reads and homozygous alternative when the alternative allele was present in more than 80% of the reads. Resulting variants were validated by Sanger sequencing. Common variants were also validated by comparing their frequency in this cohort to the frequency in GnomAD v2 [28].

Following data analysis, variants were tested for association with the otosclerosis phenotype using vtools software package (Peng B., https://github.com/vatlab/varianttools, accessed on 23 November 2022) [29]. The analysis included both the association of individual variants to the disease phenotype, as well as the cumulative effect of multiple (rare) variants.

First, associations of single variants with a minor allele frequency (MAF) above 0.01 were tested using Fisher’s Exact Test. Resulting *p*-values were corrected for multiple testing using False Discovery Rate (FDR) analysis, as implemented in the q-value package [30]. For all significant variants, the odds ratio (OR) and the MAF were calculated in the total population. OR and the 95% confidence interval (CI) were calculated in the six subpopulations and plotted using GraphPad Prism 9.4.1 (GraphPad Software, La Jolla, California USA). Significant variants were included in a conditional logistic regression analysis, testing for independent association models through backward modelling using R version 4.2.1 (R Foundation, Vienna, Austria). Linkage disequilibrium (LD) between variants was calculated using the Linkage Disequilibrium Calculator of Ensembl (EMBL-EBI, Hinxton, UK) using data from the 1000 Genomes Project database [31].

Second, gene-based tests were performed including three mutation burden tests (Combined and Multivariate collapsing test (CMC), kernel-based adaptive cluster (KBAC) test, and the Variable Thresholds method (VT)) and a variance component analysis (cAlpha test). All four gene-based tests were performed twice, using variants with a MAF in the control population below either 0.01 or 0.001. Furthermore, separate tests were carried out by variant types: (i) exonic variants, (ii) nonsynonymous and frame shift variants, (iii) intronic variants, (iv) 5′ UTR variants and (v) 3′ UTR variants.

## 3. Results

### 3.1. Sample Collection and Gene Selection

To replicate the previously reported associations from the GWAS reported by Rämö et al. [18], a total of 1696 otosclerosis cases and 1584 controls were reused from previous studies, provided by eight centers to investigate the genetic causes of otosclerosis (Appendix A). Cases and controls were matched for age, ethnicity and gender. Coding regions from *AHSG*, *MARK3*, *LINC01482*, *EYA2*, *SUPT3H*, *TGFβ1* and *RELN* were captured using smMIPs and sequenced. Resulting variants were tested for association with the phenotype.

### 3.2. Single Variant Tests Show Significant Results in Several Genes

Several of the tested genes contain single variants significantly associated with the phenotype (Table 1). A total of 18 variants in six different genes show a nominally significant association to otosclerosis. The QQ plot (Figure 1), comparing the observed distribution of the *p*-values in this study versus the distribution of *p*-values under the null hypothesis of no association, shows a strong enrichment of significant *p*-values. Correction of the *p*-values by false discovery rate analysis showed that 13 variants in five genes remained significant (Table 1), with the lowest q-value being 3.40 × 10^−13^ (Table 1). Of these variants, three were located in the *MARK3* gene, four in *LINC01482*, three in *AHSG*, two in *SUPT3H* and one in *RELN*. Variants were found in both exonic regions and the 3′ and 5′ UTR. Two variants in *TGFβ1* showed nominally significant *p*-values (0.022 and 0.31) but did not remain significant after multiple testing correction. Variants in *EYA2* did not show any significant association to otosclerosis.

To exclude false positive signals due to sequencing artifacts, significant variants from NGS data were resequenced using Sanger sequencing. All variants were confirmed (results not shown). In addition, the minor allele frequency (MAF) of all significant variants was calculated in the control group and compared to the MAF reported in the European, non-Finnish population in the GnomAD v2 database.

The presence of several associated variants in one gene can be either separate association signals or due to one underlying causing variant in LD with surrounding associated variants. To distinguish between these two possibilities, LD and conditional logistic regression analysis were performed. LD analysis showed that in at least two of the genes high LD could explain the co-occurrence of several variants (Appendix A). Logistic regression analysis showed that not all variants reached significance across all genes. In more detail, all three variants in *AHSG* showed a high r^2^ and D’ value (respectively higher than 0.77 and higher than 0.99). Conditional regression analysis in *AHSG* confirmed these results as two out of three signals remained in the final model. One of these signals was only marginally significant (*p*-value of 0.0447), suggesting that all three variants are probably attributable to one significant signal. In *MARK3* two out of three variants (rs11541718 and rs13987) were in high LD (r^2^ and D’ value of 1). After logistic regression analysis, two out of three signals remained in the final model. Other variants in other genes showed no regions of high LD and all variants remained as individual signals in the final model of the logistic regression analysis. The associated variants probably represent distinct underlying causative variants.

Consistent association was studied for all 13 significant variants across the six subpopulations. The allelic odds ratio (OR) and the 95% confidence interval were plotted for each subpopulation and the total population (Figure 2). When the OR across all subpopulations was lower (or higher) than one, association was considered consistent, which was the case in five variants (rs2273699, rs11868207, rs3744501, rs2278445, rs2229862). Four variants showed a near consistent association (rs13987, rs11541718, rs34216978, rs9369514). Variants in high LD show similar results across all subpopulations (e.g., rs4918, rs4918 and rs1071592).

Although all genes analyzed in the current study were previously reported by Rämö et al. [18], the individual variants analyzed in the GWAS are not necessarily the same as the ones identified using NGS in the current study. Therefore, to further test for replication, variants found in the current study were compared to variants found in the GWAS [18] and LD was calculated between all variants. This showed that variant rs4917 was the associated variant in *AHSG* in the GWAS and was also present in the current data. In *LINC01482* rs11868207 was found to be associated and in high LD with rs8070086, which was found in the GWAS, most likely representing the same association signal. A similar result was found in *MARK3* where one variant identified in the current study (rs2273699) is in high LD with a variant previously identified in the GWAS (rs1951391).

### 3.3. Gene-Based Tests Show No Cumulative Effect of Rare Variants

The previously mentioned single variant association test only has sufficient power when variants are present in higher frequencies (MAF > 0.01). To test the cumulative effect on the phenotype of multiple rare (MAF < 0.01) or even very rare (MAF < 0.001) variants within a gene, mutation burden and variance component tests were performed as described in the methods. Either all variants were included (with a MAF below the aforementioned cutoffs) or variants were split by variant type. Unlike the results from the single SNPs, only marginally significant associations were observed. The distribution of all *p*-values is presented in the QQ plot (Figure 1) in a similar way as the single variants, but limited diversion from the null hypothesis was observed. After correcting for the number of tests carried out, no significant associations remained (Appendix A).

## 4. Discussion

Despite the prevalence of otosclerosis, knowledge on its genetic background remains scarce. So far, only two GWAS have been performed. The first GWAS was the result of pooled samples which led to a loss of power. The recent study of Rämö et al. [18] represents a breakthrough for otosclerosis research, as it resulted in the genome-wide significance of 18 loci. The current study aimed to replicate the results of that GWAS and to perform a targeted resequencing to identify novel variants within the associated genes. Seven genes were selected and resequenced using smMIPs. The results show 13 variants within five genes (*MARK3*, *LINC01482*, *AHSG*, *SUPT3H* and *RELN*) that are significantly associated with the phenotype. Nine out of 13 significant variants show a (near) consistent association across all subpopulations. Interestingly, the effect allele of eight variants is more frequent in controls, whereas the effect allele in the other five variants is more frequent in the cases. This suggest both protective and disease-causing variants are associated with otosclerosis. These findings are in line with previous studies, where also protective and disease-causing variants have been described in otosclerosis [16,19,32]. In the two other genes, *EYA2* and *TGFβ1*, no variants were found that were significantly associated with otosclerosis after multiple testing correction. Burden testing showed no significant association in any of the interrogated candidate genes, indicating that the association found in all five genes is mainly driven by common variants. It should however be noted that (very) rare variants in *EYA2* showed a marginally significant signal in the gene-based tests. These results were not considered significant after multiple testing correction and are therefore not conclusive.

Almost all associated genes play a role in bone metabolism or mineralization or have been previously associated with other bone diseases or otosclerosis itself. *AHSG* codes for Alpha 2-Heremans Schmid Glycoprotein, also known as Fetuin-A, which is a plasma protein produced in the liver and involved in mineralization and bone metabolism [33,34]. *AHSG* can bind both *TGFβ1* and *BMP*, two genes reported to be associated with otosclerosis [21,22,32,35], and inhibit their activities [36]. *AHSG* was also identified as a novel candidate gene in a hearing loss screening in a cohort of 3006 mouse knockout strains. In this current study, the same variant (rs4917) that was associated in the GWAS [18] was replicated in this cohort.

*LINC01482* is a long intergenic non-coding RNA, which can have different functions, such as remodeling chromatin and genome architecture, RNA stabilization and transcription regulation, including enhancer-associated activity [37]. It is not yet clear how *LINC01482* specifically interferes with other genes or pathways. Variants in *LINC01482* have been associated in a GWAS with heel bone mineral density [38], but a clear role has not been described. The associated variants in this current study have not been associated with other diseases. One of the associated variants (rs11868207) was found to high LD with a variant from the otosclerosis GWAS (rs8070086) [18].

*MARK3* codes for Microtubule affinity regulating kinase 3 which plays a role in the phosphorylation of microtubule associated proteins (MAPs). In previous studies, *MARK3* has been associated with bone mineral density (BMD), an indicator for osteoporosis [39,40,41]. One of the associated variants found in this current study (rs2273699) showed a high LD with a variant found in the GWAS (rs1951391), suggesting that the same signal was identified, and previous findings were replicated in this study.

*RELN* was first associated with otosclerosis after performing a pooled GWAS [42] and replicated in several studies [18,19,43,44]. *RELN* codes for an extracellular matrix protein, Reelin, which is essential for brain development and synaptic plasticity [42,45]. Because of its function *RELN* has been associated with diseases such as bipolar disease, schizophrenia and autism [19]. In contrast to these diseases, the function of the gene is much more difficult to relate to the pathophysiology of otosclerosis. The replication of *RELN* association studies in otosclerosis, the recent GWAS and again in this current study, proves the role of the gene in the disease. In addition, expression of *RELN* was found in the stapes footplate and inner ear [42]. In this current study one variant was found to be associated. Remarkably, previous associated variants all lie within intron 2, 3 or 4, while the variant in this study is found in exon 53 and therefore no strong LD was found between this variant and previously reported variants. Since the current study focused on exonic regions, the regions harboring the previously reported intronic SNPs were not investigated and no conclusion can be drawn about the replication of *RELN*.

*SUPT3H* codes for a chromatin remodeling protein. The promotor of the gene is a regulator of *RUNX2*, a transcription factor involved in osteoblast differentiation [46]. *RUNX2* plays a role in bone formation and is also regulated by *TGFβ1* and *BMP* [47], which have been associated with otosclerosis [21,22,32,35]. *SUPT3H* and *RUNX2* were found to be associated with osteoarthritis [46,48].

*TGFβ1* has been reported in otosclerosis in previous studies [18,22,32,35]. In this study two variants were found with a nominally significant *p*-value, however, after FDR correction they did not show a significant association. One of the variants included T263I (rs1800472), which has been previously reported in association with otosclerosis in a Belgian-Dutch sample set [32]. The variant was reported to be protective against the disease as it was under-represented in otosclerosis patients compared to controls. This conclusion is confirmed in this current study where the variant is also more present in controls compared to cases. In addition to the two variants in *TGFβ1*, variants were found in two other genes that play a role in the *TGFβ1*-pathway. As mentioned above, *AHSG* can bind *TGFβ1* and *BMP* genes and inhibit their activities. In addition, *SUPT3H* plays a role in regulation *RUNX2* which is also regulated by *TGFβ1* and *BMP*. Previous studies have shown association between both *TGFβ1* and *BMP* and otosclerosis [21,22,32,35]. With the addition of two more genes, the hypothesis rises that the *TGFβ1*-pathway is involved in the onset of otosclerosis. This hypothesis is confirmed by the results of a gene-set analysis of the GWAS data, where five gene sets were significantly enriched, including Transforming Growth Factor Beta Receptor Activity Type I [18]. Transforming growth factor β1 (*TGFβ1*) is an important signaling molecule which regulates various cellular processes comprising proliferation, differentiation and migration [32]. The gene also plays an important role in both bone formation and bone resorption [47]. Variants in the transforming growth factor β (*TGF-β*) superfamily signaling pathways have been reported in a wide range of diseases, including cardiovascular diseases, skeletal abnormalities, connective tissue diseases and different types of cancer [49]. How the gene or the complete pathway influences the otic capsule specifically remains unknown. Functional studies on expression data of this pathway could be a good follow up study to further unravel the disease onset of otosclerosis.

To reduce the number of smMIPs and increase the number of studied genes, only exonic regions, 3′ and 5′ UTR were studied. The advantage of including more genes, is that this study could show a replication for five different genes. A disadvantage could be that important variants were missed. Apart from exonic variants, a few intronic variants were also found in this study. This is due to the design of the smMIPs where the overhang between exons and introns was also targeted. In previous (genome wide) association studies, many associated variants were intronic and intergenic variants. This shows the complex architecture of otosclerosis, where both coding and non-coding variants with different frequencies could play a role in the disease onset. Future research in these associated genes could also include intronic regions to accomplish a complete view on all variants associated with otosclerosis.

Association studies into complex diseases often suffer from lack of replication, with variants initially identified in one population failing to replicate in other studies. The importance of the current study lies within the independent and consistent replication of several previously reported association signals for otosclerosis. Seven genes close to significant variants from the GWAS of Rämö et al. [18] were resequenced using smMIPs. After single variant analysis, 13 variants in five genes were found significantly associated with otosclerosis. In all five genes individual single variants are associated with the phenotype, pointing to a strong contribution of common variants on the phenotype. In addition, variants were consistently associated across six subpopulations. Gene-based tests, however, did not show any significant results. There is no evidence for any cumulative effect of very rare variants in these genes.

## Figures and Tables

**Figure 1 jcm-11-06978-f001:**
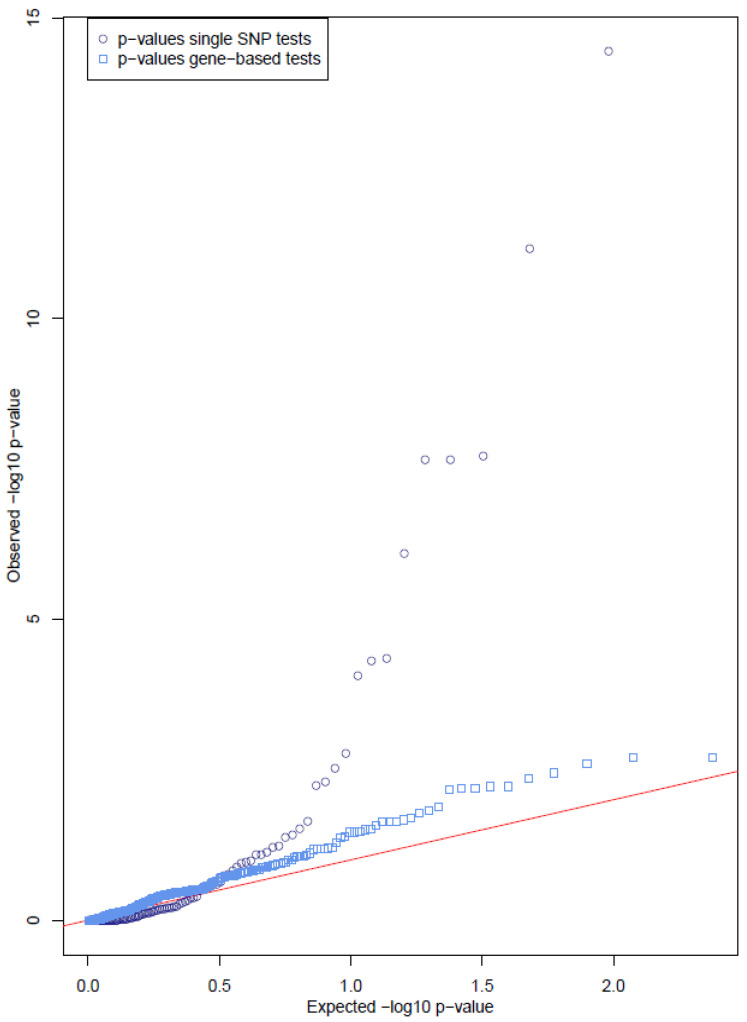
QQ plot of results of single SNP variant analysis and gene-based analysis. The QQ plot shows the observed and expected distribution of p-values of all variants used for single variant analysis. The diagonal line shows the expected distribution in absence of any association and the dots and squares represent the observed *p*-values. The QQ plot of the single SNP variant analysis (dots) shows a clear diversion of the dots compared to the red line, which indicates an enrichment of significant *p*-values. The observed p-values after gene-based analysis (squares) show only a slight diversion, indicating that there is probably no cumulative effect of (very) rare variants on the phenotype. This observation is in line with the lack of significance upon correcting the p-values for the number of tests.

**Figure 2 jcm-11-06978-f002:**
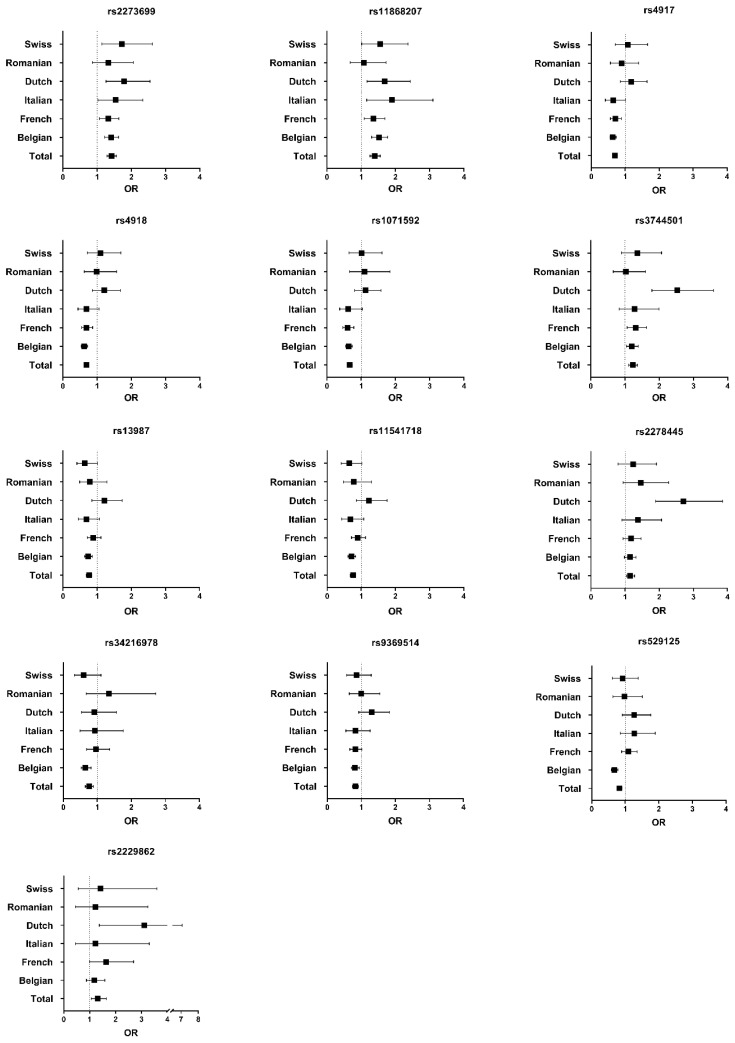
Forest plot of odds ratios of all 13 significant variants across all subpopulations and the total population. For each variant the odds ratio (OR) is represented by a square and the 95% confidence interval by a horizontal line. Association is considered to be consistent across all populations when the OR is lower (or higher) than one across all subpopulations. The plot shows a (near) consistent association in 9 out of 13 variants.

**Table 1 jcm-11-06978-t001:** Results of single SNP variant analysis. (The table gives an overview of all 13 significant variants after single SNP analysis with the uncorrected *p*-value and the corresponding q-value after false-discovery rate analysis. For each variant the corresponding rsID is given, together with the corresponding gene, region (intronic, exonic, 3′-UTR, 5′-UTR or ncRNA_exonic), cDNA change and amino acid change. Positions are given according to NM_001128918.3 (*MARK3*), NR_110825.1 (*LINC01482*), NM_001622.4 (*AHSG*), NM_003599.4 (*SUPT3H*) and NM_005045.4 (*RELN*). The number of effect alleles found in cases and in controls and the number of total alleles after quality control are given. Minor Allele Frequency (MAF) are given for the European, non-Finnish population, based on GnomAD v2 database. For each variant, the allelic odds ratio (OR) and the 95% confidence interval (CI) was calculated based on the number of effect and non-effect alleles in cases and controls).

Variant (rsID)	Gene	Region	Effect Allele in Cases/ Total Alleles in Cases	Effect Allele in Controls/Total Alleles in Controls	cDNA Change	Amino Acid Change	MAF GnomAD	Uncorrected *p*-Value	q-Value after FDR	Allelic OR (95% CI)
rs2273699	*MARK3*	intronic	1508/3210	1147/3086	c.413-4A > G	p.(=)	0.36	3.58 × 10^−15^	3.40 × 10^−13^	1.42 (1.28–1.57)
rs11868207	*LINC01482*	ncRNA_exonic	1151/3254	853/3114	n.2589T > C	/	0.25	6.87 × 10^−12^	3.26 × 10^−10^	1.40 (1.26–1.55)
rs4917	*AHSG*	exonic	1965/3260	2096/3126	c.743T > C	p.(Met249Thr)	0.66	1.90 × 10^−8^	4.25 × 10^−7^	0.70 (0.63–0.77)
rs4918	*AHSG*	exonic	1915/3202	2055/3084	c.767G > C	p.(Ser257Thr)	0.66	2.16 × 10^−8^	4.25 × 10^−7^	0.69 (0.62–0.76)
rs1071592	*AHSG*	exonic	2283/3234	2382/3102	c.810A > C	p.(=)	0.75	2.23 × 10^−8^	4.25 × 10^−7^	0.66 (0.59–0.74)
rs3744501	*LINC01482*	ncRNA_exonic	1360/3254	1116/3122	n.95A > C	/	0.33	7.96 × 10^−7^	1.26 × 10^−5^	1.23 (1.11–1.36)
rs13987	*MARK3*	UTR3	940/3220	1055/3104	c.*345=	/	0.35	4.35 × 10^−5^	0.00057	0.76 (0.68–0.85)
rs11541718	*MARK3*	UTR5	912/3126	1028/3022	c.-203=	/	0.35	4.84 × 10^−5^	0.00057	0.76 (0.68–0.84)
rs2278445	*LINC01482*	ncRNA_exonic	2293/3256	2046/3108	n.332T > G	/	0.67	8.49 × 10^−5^	0.00090	1.15 (1.04–1.28)
rs34216978	*LINC01482*	ncRNA_exonic	326/3260	391/3128	n.166A > G	/	0.14	0.0017	0.016	0.76 (0.65–0.88)
rs9369514	*SUPT3H*	UTR3	1224/3256	1284/3112	c.*282=	/	0.42	0.0029	0.025	0.82 (0.74–0.91)
rs529125	*SUPT3H*	UTR3	1691/3260	1728/3120	c.*465=	/	0.70	0.0049	0.039	0.82 (0.75–0.91)
rs2229862	*RELN*	exonic	204/3270	145/3114	c.8508C > T	p.(=)	0.052	0.0058	0.043	1.32 (1.06–1.64)

## Data Availability

The datasets generated and/or analyzed during the current study are available in the European Genome-phenome Archive (EGAS00001006792).

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
