# Peer review of "Targeted Resequencing of Otosclerosis Patients from Different Populations Replicates Results from a Previous Genome-Wide Association Study"

_jcm, 2022, doi:10.3390/jcm11236978_

Round 1
Reviewer 1 Report
Dear Corresponding Author,
I felt that the article is very well written but it could use some improvement concerning description of methods for example in patient selection. Also, I felt that some explanation about genome-wide association studies (GWAS) was necessary, since the reader could be new to this type of research. Some more empashis could be given to the conclusion, as " this study shows a replication of the results of the GWAS by Rämö et 341 al. " diminishes the importance of the presente study.
Reviewer 2 Report
This study is the replicatin study of the recent GWAS data regarding otosclerosis. The authors selected 7 genes from the previous studies and found that 18 variants among 5 genes are significantly associated with the phenotype. They did not find cumulative effects of rare variants of each genes. Overall, the results and methods are clearly described. The outcomes are confirmative of the previous data, but it is worthwhile given the importance of replication data of GWAS analysis. Several concerns are followed.
Results
1. L154; Demographic details and comparisons between control and patient groups should be described as a format of table.
2. L160; 18 variants in 5 genes, not six genes
3. L168; p-value of TGF1b variant needs to be described in the text.
4.Table 1; The information about 18 variants are not sufficient. Effect allele/noneffect allele would be better and more clear than alternative/wild allele given that some effectalleles are not minor allele as in rs4917. Genomic location, nucleotide of alleles, protrein change if applicable, MAF in the same ethnics from public DB (genomad) could be included.
5. LD data can be additionally provided as a table. Insuffcient values were described in the result.
Discussion
1. L260; data regarding EYA2 in gene-based test are not included in the main text. It needs to be added (at least in the supplemental data).
2. L265; Full name of all abbreviations should be used at the first use. It needs to be revised through the manuscript.
3. L280; Reference is missed.
4. Because this study focused on the exonic regions of interest genes, causative variants can be functionally devided into preventive and risk alleles. Based on the ORs, it can be speculated in a certain way. It is necessary to describe the plausible effects of gain/loss of function of individual genes on otosclerosis.
Round 2
Reviewer 2 Report
All the raised issues have been addressed.